# Transdisciplinary Research on Indoor Environment and Health as a Social Process

**DOI:** 10.3390/ijerph18084379

**Published:** 2021-04-20

**Authors:** Kristian Stålne, Eja Pedersen

**Affiliations:** 1Department of Materials Science and Applied Mathematics, Faculty of Technology and Society, Malmö University, Nordenskiöldsgatan 1, 205 06 Malmö, Sweden; 2Department of Architecture and Built Environment, Faculty of Engineering, Lund University, P.O. Box 118, 221 00 Lund, Sweden; eja.pedersen@arkitektur.lth.se

**Keywords:** transdisciplinarity, intradisciplinary, collective intelligence, collective leadership, collective learning, safety, indoor environments, health, well-being

## Abstract

Although issues concerning indoor environments and their interaction with humans span many disciplines, such as aerosol technology, environmental psychology, health, and building physics, they are often studied separately. This study describes a research project with the transdisciplinary aim of bridging such disciplinary boundaries. Semi-structured interviews were conducted with the twelve project members to explore their understanding of transdisciplinarity regarding the conceptual as well as social aspects of collective learning and leadership and the measures taken to achieve this. The interviews were coded in NVivo (QSR International, Doncaster, Australia), which was used to identify themes concerning notions associated with transdisciplinarity, collective leadership, collective intelligence, and learning. A shared understanding of transdisciplinarity meant that the researchers transcended their disciplinary boundaries by moving into each other’s fields. This collective learning process was facilitated by introductory lectures on each other’s fields, contributing to collective leadership and a safe atmosphere. We argue that a transdisciplinary approach is appropriate in order to address indoor environment issues as well other complex problems, for which additional time and resources should be allocated for individual and collective learning processes.

## 1. Introduction

In ancient Greece, no distinction was made between what are, today, seen as different fields of science, and a single scholar could master everything considered to be known at the time [1,2]. This universal view persisted through the centuries, but when the development of science took off in the 19th century, it was no longer possible, and scientific disciplines were defined in terms of university departments [3]. However, as the complexity of the world was acknowledged and the social sciences were also introduced, there was a need to study objects or phenomena through collaborations between different areas of knowledge, summarized and labeled by Rousseau and Wilby [4] as multidisciplinarity (i.e., simultaneously making use of several disciplines with different approaches), cross-disciplinarity (i.e., establishing a middle ground), or interdisciplinarity (i.e., attempting to synthesize disciplines into something new). The introduction of the concept of transdisciplinarity (TD) is attributed to Jean Piaget, who, together with André Lichnerowicz and Erich Jantsch in the 1970s, called for structures and systems of disciplines to be applied in education [5]. Within this theoretical route towards TD, Basarab Nicolescu developed the following definition: “Transdisciplinarity concerns that which is at once between the disciplines, across the different disciplines, and beyond all disciplines. Its goal is the understanding of the present world, of which one of the imperatives is the unity of knowledge” [6] (p. 44). The concept, thus, has its origin in a philosophical effort to understand reality from a scientific perspective and is clearly linked to the understanding of systems. Consequently, TD was found to be applicable in education and in studies of complex societal issues, such as environmental problems and sustainable development [7], i.e., solving problems outside academia at the intersection of science, technology, and society [8]. Complexity, non-linearity, reflexivity, and mutual learning were emphasized and integrated in case study research as a realization of TD [9]. With this, the role of the researcher changed as TD research came to be characterized by shared aims rather than an emphasis on individual genius. In this more applied route, TD has become a concept that refers to studies of real-world settings or problems carried out by researchers from different disciplines or by researchers together with stakeholders outside academia. Like many appealing concepts, TD risks dilution if it is used uncritically and without the original intention to achieve something beyond what interdisciplinarity entails. We, therefore, need to reflect on what TD can mean today, how it is understood, and how TD can be realized in practice. The main body of recently published literature describes and presents how TD can be incorporated in collaborations between researchers and stakeholders outside academia [10,11,12], but reports concerning cases of TD within academia seem rarer, though they can be found [13].

Several obstacles to practical TD research have been identified. Although heterogeneity and intellectual diversity are essential within a TD research group, TD simultaneously requires a shared conceptual framework for the development of methodology [14], creating a potential for conflict if not all members are properly introduced to this more process-oriented and non-hierarchical approach [15]. In particular, collaborations between the natural and social sciences risk ending up in parallel sub-studies [16], which could be explained by traditional academic cultural differences in the view of reality (e.g., positivist vs. constructivist), methods (quantitative vs. qualitative), thinking (logical vs. dialectical), and languages [17,18,19]. For too long, the division of research into different academic disciplines has driven specialization, so that researchers in different disciplines have difficulties understanding each other since knowledge of others’ fields is often poor [20]. Researchers tend to present and discuss their research at conferences and publish in journals in their own fields, possibly forced to do so because of career and financing structures. We lack shared spaces where knowledge and insights can be shared between fields. Academia is structured around traditional research fields, with separate divisions, departments, and faculties delineating boundaries that can be difficult to overcome [21]. Also, tacit barriers that are not part of official structures or regulations can be difficult to address, barriers such as cognitive and epistemological [22] or social differences, leading to territorial disputes, unwillingness to share data, and the strong link between the researcher as a person and the research, expressed as “my research” [23,24]. It has therefore been widely recognized that TD should be considered a social process in addition to a conceptual one [25,26].

The stated obstacles encountered when coordinating multiple forms of expertise and intellectual diversity motivate discussion of, first, the notion of collective intelligence and, second, what leadership is and how it should be exercised in TD projects. Although definitions vary (for an overview, see [27]), collective intelligence can be seen as a collective competence that, according to Heylighen, occurs when “a group of initially independent agents develop a collective approach to the tackling of some shared problem that is more powerful than the approach any of them might have developed individually” [28] (p. 1); he argues that the emergence of collective intelligence is intrinsically a process of self-organization since a single individual leader would likely impose a consensus view on the other group members. Weick and Roberts understood the concept of the collective mind as “a pattern of heedful interrelations of actions in a social system” [29] (p. 357). The notion of heedful indicates that, to be considered intelligent, actions must be intentional rather than accidental. Factors contributing to a group’s overall competence (i.e., collective capability) and successful performance are, besides the cognitive intelligence of the individual members, the group’s emotional intelligence and social capital. The latter consists of network properties such as the strength and quality of group relationships and cohesion, giving rise to a type of leadership appropriate for the heedful and successful collective performance of TD projects.

Development in the leadership studies field generally displays a pattern similar to that of views of TD in that it has shifted from first focusing on leaders as individuals to viewing leadership as primarily relational and collective. The individual view is rendered in studies of the traits, behaviors, and methods of individual leaders in formal managerial roles situated in hierarchical organizations who exert leadership over subordinates to achieve certain objectives.

In the 2000s, attention shifted towards studying relationships, teams, networks, and organizational systems instead of seeing the individual as the primary source of leadership. Instead, leadership started to be seen more as an emergent property of the interactions between multiple actors with a shared collective outcome. This more collective approach to leadership research was somewhat influenced by research areas such as complexity theory [30] and constructive developmental theory [31], both of which emphasize the relational, dynamic, and process-oriented nature of leadership. This shift was also motivated by technological innovation, globalization, societal shifts, value shifts, and corresponding complex challenges and has brought about a move away from hierarchical and compartmentalized ways of organizing towards more self-organized and cross-functional teams.

One such approach to defining leadership is offered by the direction, alignment, and commitment (DAC) framework [32], which broadens the notion of leadership from a leader–follower dynamic to also encompass collective aspects. Here, a desired outcome of leadership is the development of collective capability in terms of agreeing on a shared direction in terms of goals, aims, and mission; shared alignment by organizing and coordinating knowledge and work; and shared commitment among the participants to subsume their own interests to collective interests and benefits. The production of DAC is a dynamic process in that it both requires and advances collective learning.

This article describes a process in which a team of researchers with roots in natural science, social science, and technology went from a joint interest in the complex system of human health and well-being, indoor home environments, and energy use, to collaboration with the ambition of achieving a TD understanding. The article is based on interviews with the 12 team members six years after the collaboration started, at the official final stage of a large joint research study. The objective of the interview analysis was to explore how collective leadership and learning can be seen as prerequisites for developing a collective capability that results in reaching not just across disciplines, but beyond.

## 2. Background

### 2.1. Shared Concepts

This study is based on individual interviews with 12 academic researchers anchored in diverse disciplines and with partly different organizational affiliations. The study was initiated by the group in the final phase of a four-year empirical research project out of a need to evaluate and reflect on how the group, and its constituent individuals, had functioned during the project. However, the researchers had a longer shared history. They first met within a transdisciplinary program initiated by Lund University (Lund, Sweden), i.e., the Pufendorf Institute for Advanced Studies (PIAS). PIAS allows thematic groups comprising researchers from different faculties to work together for eight months, typically one day a week, free of their ordinary duties without any counterclaims [33]. The aim of PIAS is to be “a creative forum, an incubator for new ideas and a springboard for new research initiatives” [34]. To support this aim, the thematic groups meet in a specially designed building with facilities for workshops and seminars and available staff and are offered generous funding for inviting international guests. Of the 12 researchers interviewed for this study, nine initially met in 2014 in the PIAS thematic group “Holistic Approach to Healthy Indoor Environments” [35]. The initiator, who also had the role as coordinator, started the group out of frustration with knowledge of the indoor environments where people spend most of their time. Despite considerable knowledge of individual components that determine air quality, temperature, drafts, acoustics, daylight access, and other indoor characteristics, it is still common to find buildings with problems that can impair the well-being of end-users, i.e., the building occupants. In most studies of the indoor environment, the technical systems and above mentioned components are typically studied separately, resulting in a fragmented understanding. For instance, the ventilation system and its performance are rarely related to the number of airborne particles in the indoor air or to the corresponding possible health effects. Furthermore, the understanding and behavior of the tenants in terms of, for instance, airing and cooking habits have been demonstrated to significantly affect the indoor air quality, as well as the dwelling’s energy usage. An observed lack of understanding how the regulating technical systems interact, and how people in the buildings interact with these systems, called for a holistic perspective and a multidisciplinary research group.

The thematic group was comprised of 14 members and a guest professor at the time. The first eight months were dominated by activities aimed at better understanding the constituents of the indoor environment and the overall perspective. These activities included: (i) seminars in which researchers shared theories and previous empirical results from their fields; (ii) lengthy free discussions in which everyone was allowed to participate and ask questions; and (iii) meetings with invited guests, from both academia and industry, who could advance the understanding of the overall perspective. The coordinator also greatly emphasized creating a supportive social environment that would strengthen collaboration within the group and trust between group members. From the beginning, it was clear that all members were expected to contribute, not just by giving individual lectures, but by taking responsibility for the overall theme so that a broader understanding could be achieved. Though no deliveries were expected by PIAS, several proposals for future external research funding were formulated based on the emerging shared theoretical framework. At this stage, the collaboration had been philosophical and theoretical with an emphasis on trying to solve the problem of unsatisfactory indoor environments, leaving large parts of the transdisciplinary questions unchallenged.

### 2.2. The Case: Empirical Study

An opportunity to test TD in an empirical study came a year later when one of the proposed studies was funded. The People–Environments–Indoor–Renovation–Energy (PEIRE) project focused on indoor environmental quality in multi-family rental housing built in the 1960s and 1970s. Collaborations with a municipally owned housing company, the energy-providing company, and the municipality allowed the research group to follow a large renovation project and to set up an intervention study that became the major study within the project. Ten households in the area agreed to participate, i.e., let the group enter their homes to study the indoor environmental quality, the regulators of the environment, and the tenants’ understanding and behavior. Measurements were performed three times, before and after renovation and at the one-year follow-up, all in the heating season. In this project, the nine PIAS researchers were supplemented with an additional three to supply missing competences, resulting in expertise in environmental psychology, human behavior, interaction design, universal design, building physics, building services, thermal comfort, aerosol technology, exposure assessment, acoustics, daylight, and complex thinking.

Though the PEIRE project became dominated by data collecting, the aim was still to reach a holistic understanding of how technical and human factors interacted and shaped the quality of an indoor environment. At this stage, it became clear that this required more than shared practical work and free discussions between researchers. In an attempt to support the learning process and reach a shared understanding in the group, researchers working on developing relevant theoretical frameworks based on more than one discipline, such as practice theory [36], or applying a systematic approach to metatheory building [37], were invited to give seminars. Different types of leadership, such as collective leadership, and how to shift between them were explored in a series of internal workshops intended to build group cohesion and approach the holistic view that was the main interest and outcome that the project had promised its funders, in addition to the individual results of the extensive measurements.

### 2.3. Observations of the Transfer into an Empirical Study

Despite the good opportunities for learning about and developing understandings of each other’s disciplines that PIAS provided, problems arose when the group began joint empirical research, as previously described [38]. The gap between social/human disciplines, on one hand, and physical/technical disciplines, on the other, in how studies are designed became apparent during proposal writing. In the former, researchers were used to choosing methods and procedures based on previously developed theories with a transparent link between theory and method, while the physical–technical researchers tended to rely on tacit basic knowledge of natural science and standards of how to perform measurements. There were also differences within the two blocs that made it difficult to agree on best practices for the empirical measurements, leading to compromises and possibly an excessive number of measured parameters (the latter will be clear when the holistic analyses are completed). Furthermore, the fact that the study included the residents, being conducted in their homes while they were living there, meant that the study needed approval from the Central Ethical Review Board, resulting in an extensive process that required detailed descriptions of the four-year research activities and that strict ethical praxis be followed. The researchers’ varying experiences of human subject research conducted in real environmental settings instead of the laboratory were unforeseen by the project leader, and critical situations had to be addressed as they occurred. Other concerns that had to be discussed were how and where to store the large datasets resulting from the measurements, publication authorship, and how to handle divided loyalties between PEIRE and the customary intradisciplinary research groups.

The diverse views of mainly practical issues that became evident in the shift from the conceptual to the empirical phase had escaped previous attempts to reduce the gaps between research fields. However, when the interviews for the present study took place, the researchers had almost four years of experience of joint empirical studies, possibly influencing their understanding of transdisciplinary research.

## 3. Method

### 3.1. Study Group and Procedure

The idea of a more systematic attempt to describe the process that the group and individual researchers underwent in the PEIRE project was presented at a joint meeting by the present authors. In this group, K.S. was the expert on adult development and complex thinking, and E.P. was the project manager. All researchers agreed to be individually interviewed. The first interview, K.S. interviewing E.P., functioned as a pilot study. As K.S. was also a member of the group, he was interviewed by an external researcher. These two interviews are not quoted in the “Results” section but are used as background for the discussion. The interviews with the remaining 10 group members were all carried out by K.S., typically in the interviewees’ offices, and lasted 30–45 min. The group comprised five women and seven men aged 26–63 years. All had earned doctoral degrees between 7 and 29 years previously (median: 13 years), except the youngest, a PhD student at the time, and another who was conducting post-doctoral research. Most of the researchers had been in academia for their entire careers, though three also had experience in industry.

### 3.2. Interview Guide

The interviews were semi-structured [39] with a focus on the project and its transdisciplinary approach. The questions focused the discussion on previously described conceptual and social aspects of TD, aspects of process, learning from individual and collective perspectives, and how leadership is exerted from the individual and collective perspectives. The interview started with questions about participation in the PEIRE project and the interviewees’ experience of transdisciplinary research in general (Table 1). Participants were then interviewed about their perception of the PEIRE project’s approach in terms of more technical and scientific aspects as well as social and learning aspects and the methods or practices that had been valuable. They were also asked to contrast their descriptions with experiences and understanding of intradisciplinary work as well as whether the project had led to any development of the researcher in terms of learning or personal development. The interviewees were encouraged to talk freely, allowing new aspects to arise.

### 3.3. Analysis

The interviews were recorded and professionally transcribed. The text was coded by K.S. in NVivo 12 software (QSR International, Doncaster, Australia). The coding was primarily based on four categories central to the discourse of TD in relation to the PEIRE project and the involved researchers. The categories were chosen to address the previously described central aspects of TD, including the development of individual and collective learning and leadership, the participants’ ways of conceptualizing TD, and specific and practical experiences of the PEIRE project. Leadership focused on the researchers’ understanding of TD in general, on their experiences of TD in relation to the joint research project to capture methodological and practical aspects, and then on the researchers from individual and collective perspectives to capture aspects concerning prerequisite experiences, knowledge, attitudes, leadership, learning, and development (Table 2).

The final coding scheme used for the analyses was comprised of the following categories: 1. descriptions of intra- and transdisciplinarity in general; 2. descriptions of the PEIRE project in particular; 3. individual aspects of TD; and 4. social aspects of collective learning and leadership. Each category contained four to six codes reflecting themes articulated and identified in the interviews. Some codes overlap, so some quotations were categorized in more than one theme; for instance, some descriptions of TD in general intersect with descriptions specific to the PEIRE project. The coded quotations were then analyzed by both authors in an iterative process in which upcoming themes were compared with previous understandings according to the perspectives introduced earlier in this article [40].

## 4. Results

The interviews reflect experiences and lessons learned from TD and from the PEIRE project and the degree to which the PEIRE project can be regarded as TD research with identified obstacles, necessary preconditions, and steps to ensure these preconditions. A central aspect of TD was said to be not staying in one’s own field, conceptual framework, or research paradigm but also learning about and engaging in the research fields of one’s colleagues. Ways to do this cover individual aspects such as having a positive attitude towards TD research, willingness to widen one’s perspective and learn about new fields, and willingness to introduce one’s own field to others. A common description was that the individual researcher should contribute to an open and respectful atmosphere that is collegial rather than competitive. From a collective perspective, the group needs a sufficiently shared understanding (or sufficiently overlapping understandings) of the study objects, and of how to practically cooperate and address the research questions. In the research process, several obstacles were identified, ranging from using differing language, concepts, data formats, and views of how to address the research questions, to structural aspects, such as upholding a shared identity over physical distances and dealing with various organizational belongings and career incentives that favor intradisciplinarity (ID) rather than TD. Several interviewees mentioned that their understanding of TD had developed due to the research and acknowledged that performing TD research takes more time than anticipated. In the following, codes will be described in more detail and illustrated with quotations.

### 4.1. Intra- versus Transdisciplinarity

The interviewees had *different descriptions of TD*, and for some, the nature of TD seemed rather abstract. A shared basic view was that TD encompassed several research paradigms, according to which researchers need to cross boundaries into each other’s research areas and “mix” their knowledge:

Transdisciplinarity is simply when we apply … mix our knowledges. … It’s not enough for us to sit in these meetings and I write an article based on my system, you write [based] on your system. But … we mix it and it becomes transdisciplinary.[Respondent Q]

One way of describing TD is in terms of complexity, with the different research areas interacting unpredictably. Some interviewees conveyed a basic understanding of TD as crossing boundaries and mixing knowledge to form a complex system in which researchers from the incoming disciplines interact. However, the study object, in this case the indoor dwelling environment in terms of both its technical and human aspects, could be seen as complex, contributing to the high complexity of TD:

If I would describe it in a word, I think it would be complex. It’s a complex research in the sense that itself, it’s hard to understand how many different fields of research can study on the same research area, let’s say. Not research area, but the same project. And it makes it complex. It’s a team consisting of different sub-teams that should interact together, not be part of its own study items, but in general discuss and interact and understand the different areas. So that makes it complex. Another … I think it’s really complex to study … I mean, the study field itself. When you do indoor environment studies, as soon as you put people and the focus on people, people are complex by themselves. So it makes it even more complicated to actually study how … If you take into account, you want to study the heat exchange of different things and you put people in everyday life, with their behaviors in it that you don’t know, that makes it complex to acquire data and to analyze this data. So I think that’s another way of complexity that it adds. [Respondent X]

One way of describing TD is by contrasting it with ID, captured by the code *Intradisciplinarity (ID)*, which contains descriptions of how ID is typically performed and contrasted with TD. This code also contains descriptions of the limitations of intradisciplinary research:

I realize that my research area is not enough to actually study complex reality. One must consider a great many different factors in order to somehow reach an understanding of reality. There is a huge limitation in only looking at my own subject area, then you miss several other factors that affect our complex reality. [Respondent T]

ID was typically described by the interviewees as starting from well-defined expertise and a research group that shares the same knowledge and skills, and then using existing methods in the field to study well-defined problems with good precision:

Well, I think interdisciplinary research is the most common type of research. You have your research group, you have an expertise and you do the research within that expertise, and you have a very sharp … it’s easier to identify the problems, it’s easier to identify the directions, and once you’ve worked more in this field, then you get the experience that you can actually pinpoint specific problems and areas to research on and still deepen even more, so it gets even sharper and sharper and sharper. In that way, that is good, to advance in a deep area of research. In another way, one can easily ignore a lot of other interactions that happen in this complex world, especially when you include research on people, or with people. [Respondent X]

The interviewee continued by describing TD in terms of wider scope and doubts, and as entailing *difficulties and obstacles*, such as practical difficulties in studying and measuring conditions in the same apartment using many different methods simultaneously and coordinating efforts among many researchers. This also entails structural obstacles in that the academic system is, in many ways, designed for ID and for the tendencies of researchers to stay within their own fields and, for various reasons, not to interact with other fields, in what is often called “working in silos”. Moreover, TD does not give as much academic credit to the individual researcher:

So when you have a cross-disciplinary team, where each part of the team brings its own expertise and research in a deep level in certain areas, and then try to discuss it and combine the results, then you get a much more wider and you can cover more complex problems in a wider perspective. And transdisciplinary research, I think it’s very complex in a way that it requires complex processes to work together. And it’s very hard to do. I think we tried, and I’m not so sure we actually done it. But once you do it from the beginning, that you actually attack the same problems, not different problems on the same project, but the same problems from multiple points of view, then you can actually have a much better understanding on the actual complexities and solutions that could arise on the problem that you are attacking. But I don’t know how can this work in practice? I mean, it’s very hard to do as a researcher, as an academic with multiple things to do in a short time to do a project that probably need a different kind of approach. [Respondent X]

One central aspect that was given an individual code was differing perspectives, in which different views of performing research are contrasted across fields, such as: performing measurements in a controlled lab environment versus in situ; more theory-based fields (e.g., building physics) versus more empirically driven fields (e.g., aerosol technology or environmental psychology); differing views of what constitutes a sufficiently large study sample; and whether the goal is to produce accurate and generalizable models of the study object or, rather, to produce useful frameworks for interpreting it from different perspectives:

There is a clear difference in that we have … due to the presence of, yes, mainly NN, but also the others in building design who have a more quantitative approach to statistics than we have. We seldom have such large populations, so statistics and hypothesis testing are never of interest here since we can’t statistically prove things that way. We have a test house, we have five test houses, sometimes even ten test houses, and then we are happy. A population of ten in their world is absolutely zero—it’s not worth anything. [Respondent V]

The above descriptions capture some of the difficulties of TD but also illustrate the researchers’ reflections on their own research practices and the assumptions on which they are based. It may be easy to view research as either ID, in which everyone stays within their own perspectives, or TD, in which all perspectives are intertwined, but some respondents reflected on intermediate steps *towards TD*. For example, two researchers or sub-groups might interact or cooperate on a limited aspect of the project; in this case, one researcher who studies particles and aerosols interacted with a researcher dealing with installations and ventilation:

Okay. So, we’re doing particle measurements, and we are looking how far they are removed by the kitchen fan and the ventilation, for example. And that’s why we need data from N.N., because we measured air exchange rate and the speed of the kitchen fan. [Respondent U]

### 4.2. PEIRE Descriptions

A common idea among the interviewees was that TD provided the opportunity for more interesting *research questions*. The nature of the TD cooperation of the PEIRE project in terms of being able to answer wider and more complex research questions was perceived as meaningful in terms of contributing to a sustainable society and increased well-being:

This project makes me become very engaged in what I am doing here. You feel that you are having a positive impact on both the environment and those who live in it as well. It has some sort of … or has the potential to do that, anyway, I realized. That is something that is different … has been different from the others. [Respondent Q]

The PEIRE project included the *human as a study object*, meaning that the researchers had to take into consideration the tenants living in the apartment, not only as study objects but also in practical terms since the researchers were performing measurements in their homes. This entailed ethical approval and further concerns, which were new to some researchers, particularly those from a technical background. The interaction with people in their dwellings gave a new dimension to the research and was mainly described in positive terms but also as challenging because it required social skills:

I think working with the tenants may be what affected me the most, rather than the other researchers, who are still researchers, even though they come from different disciplines. But I have personally not worked as much with tenants before, and meeting them and talking with them and … It’s probably what has affected me the most as a human being in how you … well, talk to different people, still on the subject but … And it’s very personal being at home with people, especially people you don’t know and this … navigating this delicate situation, and they have invited us into their homes or we have intruded, and [we must] show respect for that. At the same time, you have to work in this environment, so I think that has been a very exciting part, actually. [Respondent V]

*Methods* applied in the PEIRE project comprised, as the interviewees described it, both established ways of performing technical measurements, such as measurements of lighting, noise, ventilation flow, and particles, as well as methods for facilitating or engaging in TD, such as systems dynamics modeling or metatheoretical frameworks. Although the project has a holistic and TD approach, most of the measurements performed and methods applied can be considered as ID, i.e., as established within the individual research traditions.

The empirical work in the PEIRE project was described by the interviewees in positive terms, also acknowledging that not everything had been optimal. The *positive appraisals* code contains statements that were positive towards certain aspects of the project and its implementation, such as emphasizing a respectful and curious atmosphere, good cooperation within the project, and good performance, whereas the *mistakes and lessons learned* code describes what the interviewees thought could be done better if they started the project again. The latter included being clearer about the rationale for certain measurements, making fewer types of temperature measurements, and communicating with the tenants in a clearer and more consistent way.

Yes, to some degree we have probably not been that good at motivating each other—“Why do I want to measure this?” And it is … in all circumstances, you always measure what you can measure. You are good at making a certain kind of measurement, so you want to include that in the project—that’s just pure laziness. [Respondent W]

A fairly common notion was that certain aspects of PEIRE could be classified as *not yet TD*. Although the interviews took place at the end of the project and there was an explicit desire for a holistic and TD approach, there seemed to be a shared notion that the group had work to do in order to be truly TD. Cited reasons for this shortcoming were time restrictions, leaving researchers only a fraction of their total time to spend on the PEIRE project, and habits of paying most of one’s attention to one’s own field. Other reasons mentioned correspond to those more general aspects discussed previously in the code *difficulties and obstacles to TD*:

So far the research is pretty silo based. The whole time we are talking about doing holistic analyses, we are supposed to be comparing cross-wise and multi-factorially and so on. And we have made attempts at doing that, but we haven’t really gone there yet for real, really in some rigorous way, neither quantitative nor qualitative, co-analyzing things that we really want to do in the spring. It will be really exciting. So what strikes me most is that, so far, after three years, there are still parallel silos in which we have gathered data and conducted partial analyses and presented preliminary results in different sectors. So it’s really exciting, it’s with … not with anxiety but with butterflies in the stomach that I look forward to our doing that, to getting it together. [Respondent O]

### 4.3. Individual Aspects

The interviewees expressed a strong commitment to TD and what it meant at an individual level, describing their own interest in and work on TD in the code *previous experiences of TD*, for instance, in the preceding PIAS theme. The code *attitudes towards and abilities for TD* contains the respondents’ views of what they think are appropriate prerequisites for participating in TD research. This includes curiosity and humility towards other disciplines and researchers, experience of more than one research field, being open-minded, having an ability to see things from several perspectives, motivation to engage in complex issues even though they may be difficult to pursue, and willingness to cooperate rather than compete. One interviewee especially emphasized territoriality, which, based on experience of other research groups, could destroy open discussion and thus contradict a main pillar of TD research. The good working atmosphere was partly attributed to the project leader but mainly to all researchers in the group, who created links to aspects of leadership.

The interviewees described what they had learned from working in the PEIRE project in terms of new competences in the code *learning for the individual*. Their main experiences referred to knowledge from other research fields represented in the project, which was crucial in building a shared understanding, but also practical issues, such as certain procedures for measuring, for example, lighting or noise. The lessons learned also involved secondary aspects of research, such as how to communicate with other researchers and the public in an effective way, as well as how to cooperate in practice. The latter requires that one become more aware of the structural aspects of the academic environment and overcome the administrative barriers that arise when participants are employed in different institutions or universities, such as managing finances and the rules of dividing responsibilities between management and individual researchers. The informal “meetings in the corridor” in which daily problems are solved also need to be replaced with new ways to create connections between individuals and facilitate work.

The interviewees had also experienced *personal development*, which covers wider aspects beyond simply learning new competences or knowledge. These responses typically described an increasing awareness and tolerance of others’ thoughts and disciplines and an emerging acceptance of the limited nature of one’s own field. The nature of the broad and holistic approach gave the interviewees more space as people, described by some as better awareness of how humans relate to their surroundings. These insights helped the interviewees to mature, as both researchers and individuals:

And then it is generally discussed, and I sit and reflect in the back of my mind … on what this says about either this person who is proposing something or, in general, on what is being proposed, how that fits into my general understanding of people and their relationship to the world and to flows of information/communication, scientifically as well as daily, and I get some confirmation and things add up. And it helps me mature as a human being and not only as a researcher with scientific competence, I guess. [Respondent O]

### 4.4. Social Aspects

As revealed in the above quotations, social aspects were very important in the PEIRE project. Most interviewees had been present at the PEIRE seminar in which collective leadership was explored and described as leadership that is not exerted by the formal leader only (which is also included in the code as a counterpoint) but rather by the group as a whole. Several interviewees described how they tried to contribute to this collective leadership by committing to and taking their own initiatives in doing their tasks, as well as by upholding the group cohesion and atmosphere through informal meetings and sharing resources, such as articles and new funding opportunities. Key concepts here are trust in and commitment to the notion of collective leadership:

I think it has been carried out as planned, as people expected. So that’s good. And leadership is that, at the beginning, we have this collective leadership, that’s how we make our plan, made our proposal, and make our mindset ready for this. I think we had already agreement that … how this will look like. I think it’s good to have proposal together, everybody have agreement. And then … That’s first, I think, important. We agreed we would do it, make a commitment. [Respondent R]

The interviewees’ descriptions of *cooperation* within the project revealed a clear shared understanding among the participants. It was generally acknowledged that the project, and therefore, all the involved researchers, aimed for a holistic approach and that the main focus was the measurements, results, and joint analyses in the studied apartments. To achieve this, knowledge from each field had to be transferred to other fields, and efforts had to be made to arrange meetings and organize measurements in the apartments. *Group dynamics* were addressed as a key to how the group had developed, and to how many aspects of the research basis (e.g., conceptual models and theoretical frameworks) had to emerge as a result of joint efforts rather than there being an existing theory that could simply be applied. Group dynamics also contributed to the character of the group meetings, with everyone being expected to participate rather than a few dominant individuals having the most to say. This atmosphere was described as a dynamic entity that had to be continuously maintained. Setbacks in the shared understanding and group cohesion had to be addressed when they became obvious, and new individuals coming into the group had to be introduced so that they could adapt to the group and its workings:

For me it’s really difficult to say, this is what I remember from Pufendorf, but nothing is static and nothing is … It’s not like this, that once you achieve something it will just continue, you still have to work on it, otherwise it will not continue in the same way. Also the group dynamics, as it’s called, it’s dynamic, it changes with time as well so you have to make sure that you do not allow this kind of maybe negative [laughter] things to pop up, to not encourage it anyway. But for me it’s really the … I think it’s a combination, but I think that what we gained really a lot is that we knew each other, we could understand from where we are coming from, and I think that also this clarification of our goals and what we are trying to do, which in fact it was easier because we’ve written together proposals so we had the frame for what we wanted to do, so it was easier to focus on it as well. [Respondent P]

Partly based on previous experiences of competition, the participants discussed whether they saw themselves primarily as competitors or colleagues, both of the other project participants and of other research groups. A common understanding was that the research questions were so complex that a single research group—even less so individual researchers—could not come up with sufficient answers. Therefore, it was natural to regard each other, as well as other research groups, as colleagues. This notion was also reflected in activities such as seminars and cooperation events to which other researchers and groups had been invited:

Then, it is probably also that we have all simply been lucky enough to have respect for each other, and everyone is simply quite kind, I would say. There is no one who … it doesn’t seem as if there is anyone who wants to win, but everyone just wants to do … is interested in their respective research fields, and I just think it’s fun if someone else has a different angle on it. But territorial behavior, I see none. And that can ruin, when you cannot take … yes, when it becomes territorial marking. When the comments are intended to win some kind of point and the answers are to defend some kind of territory. But here the comments are just because you think its fun to twist and turn the questions, and in a safe environment. … When people have to win, it can work anyway, but then it can falter, because then some will not speak up, that’s what happens. Then, some go quiet and input stops coming from that direction. [Respondent V]

The previous quotation also cites an example of a *safe atmosphere*, which refers to having an open and respectful discussion climate in which everyone is expected to contribute, and it is always acceptable to ask questions, even from a basic level of understanding. This atmosphere also constitutes a space where it is acceptable to disagree on important matters.

Several interviewees addressed the importance of having a *shared language*, which entails everything from using broad and ill-defined concepts such as “holistic”, “system”, “health”, and “transdisciplinarity” in a way that all agree on. Some terms had different meanings depending on the research field, such as “modeling” or “thermal comfort”:

So then, for me … that is hard for me to use that, but then I need to learn to calibrate with each other, what do we mean. But it’s okay, eventually. I can accept a change to thermal comfort, but I need to calibrate myself. This means the neutral way, it doesn’t mean it’s comfort, it means the sensation about thermal feeling, if you’re warm or if you’re cold. It’s not about comfort [laughter]. [Respondent R]

And better in communicating, I also think, to understand what the others are saying. These language barriers have decreased with time so that we understand each other better, have the same language. We more and more use the same language, or have a shared language, when we have a meeting. That was pretty frustrating, at least I think, when we were at Pufendorf at the beginning. It was very hard to communicate. [Respondent W]

Language issues also applied to data formats and writing styles, with researchers from construction engineering being described as writing in more direct, spoken-style language to appeal to practitioners. The depth in which some concepts were understood varied, with some researchers perceived as having a more popular understanding, while others had a very specific technical definition. This relates to the code *introducing research to others* and how the researchers introduced their research to their colleagues to build a shared understanding. One way was to give presentations in different contexts, both to each other and to the public, as a way to learn how to express oneself to convey one’s message:

But also trying to explain certain things, you know which are important in my disciplines, I think that you … Yeah, you learn how to formulate it so it would be a bit more approachable for others. [Respondent P]

Inviting and introducing colleagues to one’s research field is a crucial step in building a shared understanding. However, this entails a trade-off in the depth of understanding the invited colleagues need in a certain field and to what extent the introducing researcher can simplify his or her field:

Yes, exactly, that my level of understanding in some areas is up here, and then you notice that the others are lower, which is not strange because it is my research area. And then it becomes difficult when those with a lower level of understanding want to contribute and say, “Yes, but if we do this and this, we interpret it as this and this”. “No, but that’s not true because we have to consider this and this as well”. Yes, but like this, there’s frustration with someone … and it’s inevitable in a way when someone with a lower [level of] understanding and knowledge tries to contribute. But it will be like “But my God, we know that, it’s nothing new”. [Respondent T]

According to our usage of TD, researchers should not stay only in their own fields but move across disciplinary boundaries into other fields. The interviewees cited several examples of how this was practically done in the project, for example, by inviting and introducing others to one’s field to build a shared understanding of the complex area studied. The introduced researchers might even make valuable contributions to the fields they are new to, although one often underestimates the time and effort it takes to achieve the depth of understanding required to reach the research frontline of a field.

## 5. Discussion

The results illustrate the PEIRE research group’s understanding and experiences of transdisciplinary research. To summarize, the interviewees experienced an explicit intention within the group to perform TD research and not to stay within one’s disciplinary boundaries. The motivation for the individual researcher to adopt this view came from having an interest in other fields and from seeing TD as meaningful. TD was seen as addressing research questions going beyond what a single discipline could cover in terms of how different aspects of the indoor environment interrelated as a system, of which the tenant was part. To address this, the interviewees explained that it did not suffice for all the researchers to perform their research solely from their own perspectives in isolation, “staying in their bubbles”, but that they also needed to interact with each other more deeply. This approach generally entailed certain obstacles: conceptual, structural, and social. Cited conceptual obstacles were that researchers from different fields have different research perspectives and ideals and that they have languages that are unique to their fields and, thus, use terms differently, possibly causing miscommunication. Cited structural obstacles were being physically distant from each other and having to deal with organizational differences and incentive structures that may not benefit TD. Cited social aspects were that territorial issues may cause competition between researchers promoting their own views or research paradigms at the expense of others, rather than seeing the other researchers’ views and fields as complementary and disagreements more as learning opportunities.

Some methodological shortcomings should be noted. This is a case study with a limited number of interviewees, all of whom were engaged in the same research project on indoor environments and operating largely in the same academic environment. The perspectives and conclusions of this study should therefore be explored and confirmed in further studies. All codes were generated inductively from textual data, although the four categories were determined before the coding from the theoretical pre-understanding and from the aim of capturing not only conceptual but also social aspects of the research process. This is reflected in the introduction, the interview guide, and the shared understanding and aim among the PEIRE researchers.

From the interviews, we can observe that the researchers adopted different roles relative to each other. They needed to see themselves not only as experts in their own research fields, communicating with each other as one expert to another, but also as teachers of their colleagues and as students of the others’ research fields. All participants needed a basic understanding of all involved fields to achieve a shared understanding. Insights from research into collective leadership indicate that individuals need to see themselves as a collective and, thus, as contributing to a shared identity and understanding through how they interact, cooperate, and initiate various activities. Just as the study object, i.e., the indoor environment, can be understood as a complex system of interacting parts, the researchers may also see themselves as such. We argue that TD should be understood not only as a conceptual process generating outcomes in terms of new knowledge, theories, or frameworks but also as a social process in which learning and interaction occur between researchers. This understanding emerged from the interviews and served as a point of departure of the project as a whole and of this study in particular. Notions of collective intelligence and collective leadership have, therefore, been introduced and applied in the project and the study.

Although taking account of the social aspects seems challenging, several interviewees said that this had been addressed in a heedful and appropriate manner and that the PEIRE project could be characterized as having a trusting and non-competitive social environment in which the participants were willing to engage in other fields, possibly as beginners, and to invite and introduce others to their own fields. Starting in the PIAS theme, the PEIRE researchers anticipated differences in perspective and had access to conceptual resources facilitating TD research, such as metatheories; systems theories; and perspectives on collective leadership, collective intelligence, and knowledge integration. These were explicitly and heedfully integrated in the PEIRE project. However, despite all these efforts, the interviewees identified further obstacles and reasons why the research was still not as TD as they had hoped. The differences between the research perspectives became apparent when the group transitioned from the more conceptual discussions of the PIAS theme to the empirical PEIRE project once data had been gathered from different methods and fields and when systems modeling between the different fields was initiated. Here, the group needed to move from acknowledging differences in perspective towards resolving them in practical situations, and this included making in situ measurements as well as joint analyses of the results and shared conclusions.

The process of building a shared understanding and collective capability in TD research can be illustrated in Figure 1. Here, two research fields are represented by two bubbles in blue, with one researcher’s learning corresponding to moving up the bubble, from being a beginner, perhaps as a graduate student, at the bottom, to being an established expert, operating at the frontline of the research field, at the bubble’s top. A central dimension of the collective learning process is, thus, the developmental aspect of each bubble, indicated by the vertical axis in the figure, which broadly represents the level of capability in terms of complexity or depth and expertise in the field. From this illustration, it may be easy to imagine TD research as different experts communicating and cooperating from the tops of their respective bubbles to address the research question, shown in the yellow field. However, as previously described and discussed, for research to be TD, researchers must engage in their colleagues’ fields. This means moving into other researchers’ bubbles, typically starting at a lower level and then moving upwards. In a safe and non-competitive environment with shared leadership, established researchers in given fields can facilitate their colleagues’ learning, contributing to the collective’s capability. This process of collective learning and leadership in the TD research area can be viewed as creating sufficient overlap between all participants’ capabilities and is illustrated by the arrow between the fields. This is appropriately understood as something the group does as a shared effort in terms of collective learning and leadership.

The practical implications of this can be expressed as a trade-off that is particularly salient in the academic environment: How much time and resources should the researcher invest in this learning process, and how far up the new bubble must the researcher go in order to contribute to an adequate collective capability? This form of individual learning is seldom included in the research plan or budget. A further trade-off for the researcher who acts as a teacher is: How much can I allow myself to simplify my research area to support my colleagues’ learning in my field and contribute to the shared understanding without compromising the integrity and academic standard of my field? This depth dimension of each field also brings to the fore a category of miscommunication between researchers, resulting in one researcher having a deeper and more complex understanding of a concept than does another. When such miscommunication occurs, it may be misinterpreted as having a territorial or competitive basis. Instead, it can be seen as a natural learning process that should be expected to take time and resources, implying a need for funders to allot resources for the learning of the individual researchers and the science community as a whole. We need to acknowledge research that is not primarily intended to advance the leading edge of a particular discipline but has its strengths in the interaction and synthesis between fields.

## 6. Conclusions

The great societal challenges of our time call for new scientific approaches, and the introduction of TD processes is a step towards the collective scientific capability needed to address these challenges. Based on previously published discussions of the concept of and insights from this interview study, we believe that it is necessary and possible to conduct empirical research regarding indoor environment issues and other research areas that can be categorized as TD. A prerequisite is that the individual researchers have the initial will, a positive attitude, and recognize TD as beneficial for their own research as well as for going beyond it. TD should, thus, not be initiated by a manager who exerts leadership over subordinates in a hierarchical manner or follows a specified recipe.

We have identified several important conditions that should exist for TD to emerge. First, time and resources should be allocated for individual learning in the fields of others for the process of forming collective capability and for finding a balance between the two. Second, there needs to be a conscious and heedful approach to creating a work environment, in both the group and the organization, that supports collective leadership and learning. Third, the organizational and financial structures of the academy and other involved institutions should promote shared research grants, provide platforms for TD research products, and recognize collective capability and research outcomes, also as a basis for individual researchers’ careers.

## Figures and Tables

**Figure 1 ijerph-18-04379-f001:**
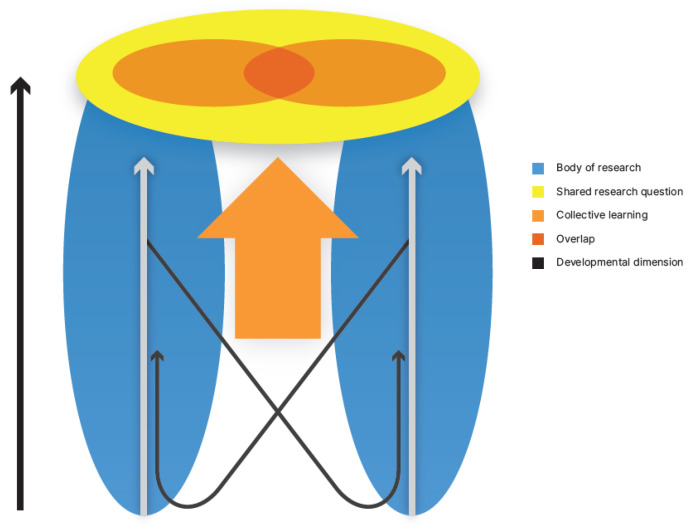
Two bodies of research addressing a shared research question. Illustration by Emily Laneryd Fujiwara. The arrow on the left is a general developmental dimension as indicated in the legend. The two thinner black arrows indicate a developmental dimension of the individual.

**Table 1 ijerph-18-04379-t001:** Interview guide. PEIRE refers to the research project People–Environments–Indoor–Renovation–Energy.

Initially, the respondent’s personal background in relation to the project was discussed: - Why are you part of the PEIRE project? What led you to this and what motivated you to be part of this project?
Then, the respondent was asked to describe the PEIRE project in contrast to other projects or ways of performing research: - How would you describe the research approach in the PEIRE project with respect to, for example, method, theory, practical aspects, and research idea? For instance, by contrasting with previous experiences of intradisciplinary, transdisciplinary research, or interdisciplinary research. Are there differences from how you perform research in other constellations? Do you see advantages or drawbacks of the approach in PEIRE? How have you practically carried out the research?
After this, the social and leadership aspects of the project were discussed: - How would you describe the nature of the cooperation between the researchers and within the PEIRE group? Are there differences, advantages, or drawbacks compared with how it is carried out in other constellations? The PEIRE project has had a holistic and transdisciplinary approach—what has been done practically to achieve this? Have there been any obstacles or difficulties?
Here, aspects of the personal development of the researchers were explored: - Have you developed your competence as a researcher from participating in the PEIRE project? Further widening the question, have you developed as a researcher and as a person from this?
Finally, the respondent’s view of what transdisciplinarity means is explored. - In the interview, we have been discussing transdisciplinarity, but how do you interpret that term? What does transdisciplinarity mean to you?

**Table 2 ijerph-18-04379-t002:** Categories and codes used in the analyses of the transcribed interviews.

Category	Code	Description
1. Intradisciplinarity (ID) vs. transdisciplinarity (TD)	Descriptions of TD	Descriptions of what transdisciplinarity means, in both general and concrete terms (citing examples), for example, by contrasting with multi- or interdisciplinarity
	Complexity	Characterizing TD in terms of complexity
	Intradisciplinarity (ID)	Describing and reflecting on intradisciplinary research in contrast to TD; limitations of ID are also addressed
	Difficulties and obstacles	Describing difficulties and obstacles in conducting transdisciplinary research with a focus on structural aspects
	Differing perspectives	Reflections on and comparisons between the different research perspectives and practices of the different fields
	Towards TD	This code describes intermediate steps from ID to TD, for example, through the interaction of two adjacent but different fields
2. Experiences of the joint research project	Research questions	Description of the research questions of the joint research project
	Humans as study objects	Descriptions of having human beings, here the tenants, be part of the study object and of interacting with them as part of the project
	Methods	Regarding the methods applied in the project, ID as well as TD
	Positive appraisals	Positive appraisals of successful elements or people in the project
	Mistakes and lessons learned	Descriptions of mistakes made in the project and what can be learned from them
	Not yet TD	Perception that work and effort are still needed for the project to be considered TD
3. Individual aspects	Previous experiences of TD	Descriptions of previous experiences of TD
	Attitudes towards and capabilities for TD	This concerns individual attitudes and capabilities argued to be necessary to successfully engage in TD research
	Learning for the individual	Descriptions of what has been learned in terms of new competences or knowledge from experiences of the project; how to learn about and get acquainted with the fields of the other participants
	Personal development	Descriptions of how the respondent has grown and developed as a person in terms of attitudes and abilities and not only in terms of competences as a researcher
4. Social aspects	Collective leadership	Descriptions of what collective leadership is and how it was manifested in the project; contrasting collective leadership to individual and formal leadership
	Cooperation	General descriptions of the nature of the cooperation within the project and the group
	Group dynamics	Descriptions of how the group changed and developed over time, and of measures to uphold a shared identity
	Competition	Description of how group members see each other and other research groups as colleagues rather than competitors
	Safe atmosphere	Descriptions of the importance of ensuring a safe and respectful atmosphere concerning each other’s expertise, and of measures to create such an atmosphere
	Shared language	Reflections on the importance of finding and using a shared language for key terms, data formats, article formats, etc.
	Introducing research to others	How to simplify, introduce, and teach research from one’s own field to participants from other fields

## Data Availability

Recorded and transcribed interviews are stored with K.S.

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
