# Peer review of "Transdisciplinary Research on Indoor Environment and Health as a Social Process"

_ijerph, 2021, doi:10.3390/ijerph18084379_

Round 1

Reviewer 1 Report

A good and interesting article that comprehensively describes the research procedure, course, results and conclusions about "how collective leadership and learning can be seen as prerequisites for developing a col-131 lective capability that results in reaching not just across disciplines, but beyond". From the substantive point of view, I have no comments or suggestions for corrections - the structure of the article is proper, the message is structured and logical, good introduction, good description of research method and how it was carried out. Interesting "discussion part" and conclusions, aim of the article is achieved. From the technical side - the title of Figure 2 should be definitely shortened, and moreover, it is the first figure (?) So it should be "Figure 1" Summarizing - congratulations to the authors. I read with interest

Author Response

Thank you for reviewing and for your remarks! Indeed, the figure is the first one and its title has been shortened since it there is an elaborate described below in the text.

Reviewer 2 Report

The perspective of this research is very interesting, deserving more attention in this post pandemic era. The insights if more refined and articulated would be constructive for narrow the gap among indoor users, policy makers, practitioners, etc. The logic of the paper is somewhat fluffy.

The following is more specific comments for consideration.

  1. More information about indoor environment, and why it need TD should be adequately presented in the Introduction. The author should also address why the investigation method and methodology in pertinent for such study, i.e., the gap in terms of TD and indoor environment in the introduction.
  1. What particular new theoretical, methodological and practical contributions the study makes need to be more clearly spelled out in the discussion. Limitation seems necessary

Author Response

Thank you for reviewing and for your remarks!

The logic and the method have been further argued for by inserting some clarifications and references regarding transdisciplinarity as a social process in the introduction as well as regarding thematic analysis in the methods section.

Some further information on indoor environment issues and the motivation for TD has also been added in the introduction.

Regarding methodological considerations and limitations, a much needed paragraph have been added in the discussion on lines 670-678.

A central consideration and practical result of the paper it that TD should be regarded as a social learning process and not only a conceptual one, which has been stated in the discussion. Even though the research group succeeds in creating a cooperative and safe environment, the learning process should be supported.

Reviewer 3 Report

It must draw a clear distinction between transdisciplinarity and interdisciplinarity. An effective definition of one and the other is important, so as not to confuse. Nature published an interesting review on the subject a few years ago. A continuation of this discussion can be reviewed in the following work https://www.nature.com/articles/s41599-019-0352-4

 The work with interviews is very well presented. While the frequency and repetition of key concepts - without altering the methodological process itself - could be modeled, the outcome and discussion raised is consistent with the purpose of the manuscript and theoretical debate.

The safe atmosphere and shared language categories are the most consistent of the study because from the perspective of the Social Sciences, the big problem facing transdisciplinarity is in two or three languages that fail to decipher each other. It is important that this discussion is further exploited in the study's findings.

Author Response

Thank you for reviewing and for your remarks!

We have added some references to further strengthen the connection to social science perspectives.

Regarding the definition of transdisciplinarity and distinction with interdisciplinarity, we consider that one important result of the study is to answer that question: “for research to be TD, researchers must engage in their colleagues’ fields” (line 727). Therefore, we refrain from giving a clear and premature definition in the article’s introduction.

Reviewer 4 Report

 This is a high quality manuscript written to an exemplary standard.

The standard and style of the writing is very high.

Author Response

Thank you for reviewing!

Reviewer 5 Report

The authors present a study about Transdisciplinary Research on Indoor Environment and Health as a Social Process. To achieve the results interviews were conducted to explore their understanding of transdisciplinary. The investigation is very interesting.

Author Response

Thank you for reviewing!

Round 2

Reviewer 2 Report

The authors have addressed my questions with revisions, and improved the manuscript. I suggest proofread it and improve the English writing. Besides, what particular new theoretical and methodological contributions the study makes still need to be more clearly presented in the discussion.

Author Response

Thank you for pointing out further improvements. Would it be possible for you to elaborate on in which way we could present our new theoretical and methodological improvements?

With regards to methodology, the study is carried out without particular novelties, using semi-structured interviews and thematical analysis in conventional way. If you refer to the entire PEIRE research project as part of the method, the novelties is introducing and applying the collective intelligence and leadership in our work. The study itself can thus be seen as part of a process of collective self-reflection among the project participants and group as a whole.

The new theoretical contributions that we make are primarily in bringing in different theoretical strands, collective intelligence and learning, and collective leadership, and applying them practically in our TD work. To advance some sort of (meta-)theoretical framework around how these strands can be combined are beyond the ambition and scope of the study. I would argue that the main theoretical novelty and implication is presented in Figure 1 which illustrates and introduces the learning dimension of adjacent research fields that one researcher needs to address in order to engage in the colleagues' respective knowledge areas. This learning dimension is central to our understanding of what TD is and how it should be engaged.

If there are other aspects that you think should be brought forth in the discussion around methodological and theoretical contributions or ways of interpret this, it would be helpful to give us some further description.

The manuscript was proofread before submission of the original submission, but not the latest additions. This can be improved.